# Evolution of the quality of prenatal care in the primary network of Brazil from 2012 to 2018: What can (and should) improve?

**Elaine Tomasi**[1,2,3]*, **Thales Moura de Assis**[4], **Paulo Guilherme Muller**[4], **Denise Silva da Silveira**[1,3], **Rosália Garcia Neves**[5], **Everton Fantinel**[1,6], **Elaine Thumé**[3,6], **Luiz Augusto Facchini**[1,2,3,6]

**1** Department of Social Medicine–Universidade Federal de Pelotas, Pelotas, Brazil, **2** Postgraduate Programme in Epidemiology, Universidade Federal de Pelotas, Pelotas, Brazil, **3** Postgraduate Programme in Family Health (PROFSAÚDE), Universidade Federal de Pelotas, Pelotas, Brazil, **4** Faculty of Medicine, Universidade Federal de Pelotas, Pelotas, Brazil, **5** State Department of Health, Rio Grande do Sul, Brazil, **6** Postgraduate Programme in Nursing, Universidade Federal de Pelotas, Pelotas, Brazil

* tomasiet@gmail.com

**Data Availability Statement:** The original data are held in a public webpage http://aps.saude.gov.br/ape/pmaq and can be located using the search

## Abstract

The article describes the temporal evolution of prenatal quality indicators in the primary health care network in Brazil and investigates regional differences. This study used data from the external evaluation of Brazil's National Program for Improving Primary Care Access and Quality (PMAQ) with health teams participating in Cycles I, II and III of the Program, carried out respectively in 2012, 2013/14 and 2017/18. The number of visits, physical examination procedures, guidelines and request for laboratory tests were investigated. There was a positive evolution for tests—HIV, syphilis, blood glucose and ultrasound, and for all tests, guidance on feeding and weight gain of the baby and examination of the oral cavity. The indicators that performed the worst were: performance of tetanus vaccine, six or more visits, receiving guidance on exclusive breastfeeding and care for the newborn, and the procedures—all, measurement of uterine height, gynecological exam and cervix cancer prevention. These changes had a varied behavior between the regions of the country.

## Introduction

Prenatal care, pioneered by Ballantyne in 1901, systematizes a set of conducts and guidelines directed at pregnant women [1] There is already evidence that women adequately monitored during pregnancy have better health indicators, as is the case, for example, of obese pregnant women or those with a family history who need to carry out the prevention and prophylaxis of pre-eclampsia [2]. Also important is the role of prophylaxis with antibiotics in urinary tract infections, which can result in maternal complications such as perinephric abscess and urinary obstruction, in addition to nutritional control in the case of gestational diabetes, responsible for premature labor [3–5]. In addition to the primary objective of early identification of possible pathologies, abnormalities or complications, pregnant women may be followed-up based on the guidelines of the United Nations (UN) and the United Nations Children's Fund (UNICEF) [6–9].

terms 'cycle 1 PMAQ microdata', 'cycle 2 PMAQ microdata' and 'cycle 3 PMAQ microdata'.

**Funding:** The data from this study were produced within the scope of an external evaluation process of the Program for Improving Access and Quality of Primary Care (PMAQ-AB). This program was fully funded by the Ministry of Health, which transferred resources to several federal universities for them to conduct fieldwork in 2012, 2013/2014 and 2017/2018. Resources were decentralized to Universities and managed by supporting foundations. In our case, the Federal University of Pelotas and the Delfim Mendes da Silveira Foundation (FDMS) and the University Support Foundation (FAU). The resources were mainly applied in tickets and daily rates for interviewers, acquisition of tablets and accessories, and research grants. Research grants were not a salary, they were a temporary incentive for researchers to dedicate themselves to the project's activities. Data are available in open access to the entire population. Our research group – and others across the country – has been producing several scientific articles, published in Brazil and abroad, since 2013. In total, the amount transferred to our institution, responsible for 20% of data collection throughout Brazil, was $3,386,415.The Ministry of Health only guided the development and standardization of data collection instruments, so that the information collected by the universities could also be used in the teams' certification process. This certification defined the values of transfers to municipalities. Funders did not participate in data analysis, decision to publish or preparation of the manuscript.

**Competing interests:** The authors have declared that no competing interest exist.

Quality in health has been assessed from the perspective of comprehensiveness or comprehensive care, through a holistic approach that considers biological, psychological and sociocultural aspects together with the provision of humanized actions for the promotion, protection, prevention, recovery and rehabilitation of health [10–13]. Comprehensiveness is one of the most relevant principles of the Unified Health System, in addition to being an important attribute in the assessment of the quality of PHC, a privileged space for its expression [11]. Problems in the completeness of health actions defined in protocols, consensuses, guides and official guidelines for specific health conditions reflect failures in the comprehensiveness of care and thus in the quality of PHC, as evidenced in studies on women's and children's health, of users with chronic conditions and elderly [14–16]. In this perspective, the assessment of the completeness of a repertoire of health actions is valuable to estimate the comprehensiveness of care and, thus, its quality [15–17].

In the scope of prenatal care, the Ministry of Health, in line with the recommendations of the World Health Organization (WHO), recommends a series of procedures to be adopted by health services and professionals in favor of maternal and child health, which are organized in the Low-Risk Prenatal Care guideline [18]. Prenatal care is one of the priority programs offered in the context of Primary Health Care (PHC), based on the principles of the Unified Health System (SUS) of universalization, equity and integrality, to ensure the constitutional right of universal and equal access to health [19, 20]. Two important programs operationalize this priority: the Prenatal and Birth Humanization Program (PHPN) and the Cegonha Network, with an emphasis on welcoming with risk and vulnerability assessment and classification, expanding access and improving the quality of prenatal care and linking the pregnant woman to the referral unit for childbirth [21]. Since 1994, when the Family Health Program was implemented in Brazil, health care has been improving, culminating in the adoption of the Family Health Strategy (FHS) as the main model of primary health care [13, 22]. Despite its growing scope, it is known that there is still much to improve in its coverage and quality. According to the 2019 National health survey [23], 70% respondents reported seeking public services when they need health care, a fact corroborated by the proportion of Brazilians who have health insurance (28.5%).

Even with advances in social and demographic indicators between 2000 and 2009, inequalities in access to prenatal care still persist, mainly in the number of visits carried out according to the age of the pregnant woman, with the worst situation for adolescents and women at the end of reproductive life [16, 24–28] In 2011, the National Program for the Improvement of Access and Quality in Primary Care (PMAQ-AB) was instituted in an evaluation process of primary health care focused on the FHS, one of the largest PHC performance assessment and incentive programs ever developed in Brazil and in the world [29–31]. The PMAQ was not characterized as a pay-for-performance program directly to health professionals. The incentive to improve access and quality was associated with a complex certification, which included the self-assessment of professionals and managers, selected indicators from secondary sources and external assessment, resulting in the financial value to be transferred from the federal management to the municipalities.Currently, PMAQ has completed three cycles, obtaining information on various indicators of quality of health care in primary network, including prenatal care, however, publications on Cycle III and comparisons of the three assessment cycles are still rare [32]. Data from PMAQ Cycles I and II for the whole of Brazil revealed important deficits and inequalities in quality indicators [16, 17]. In addition, there is evidence of sociodemographic inequities in access to prenatal care, with greater restrictions for young, black women with low education [33]. Since December 2019, a new financing model for the Brazilian APS has been instituted, the Prevent Brazil Program [34].

The objective of this article was to describe the temporal evolution of the coverage and quality of prenatal care performed in primary care in Brazil over the three cycles of the external assessment of PMAQ, and to investigate regional differences.

## Materials and methods

This was a cross-sectional design and was an excerpt from the external assessment phase of PMAQ, with health teams participating in Cycles I, II and III of the Program, carried out respectively in 2012, 2013/14 and 2017/18. Studies were conducted under the coordination of 41 Brazilian Higher Education institutions, led by the Oswaldo Cruz Foundation (Fiocruz), Federal University of Bahia (UFBA), Federal University of Minas Gerais (UFMG), Federal University of Pelotas (UFPel), Federal University of Piauí (UFPI), Federal University of Rio Grande do Sul (UFRGS), Federal University of Rio Grande do Norte (UFRN) and Federal University of Sergipe (UFS) in partnership with the Department of Primary Care of the Ministry of Health.

Data were collected by trained interviewers, using electronic questionnaires using electronic tablets. The instruments consisted of three modules common to the cycles: I—observation in the basic unit, with questions about service infrastructure; II—interview with a higher education professional about the team work process; III—interview with users waiting for service at the health unit on the day of the assessment. Users aged 18 or over were interviewed and those who were either at the UBS for the first time or who had been treated for more than 12 months were excluded. The interviews were conducted in the UBS waiting room, before medical appointments, and women who had children under two years of age were invited to participate.

There was a pilot study before each data collection in order to test the instruments and the logistics of the field work. There was no sample size calculation as this was a study based on services and teams that voluntarily enrolled in the program. It is not possible to estimate the proportion of the eligible population that this sample represents in each of the districts; it was not the intention of the study to make inferences for each location, so representativeness was not a central issue; it is understood that the users interviewed, being frequent at the UBS, could provide reliable information about the care provided by the teams.

Data collection carried out by the 41 institutions was organized by eight leading institutions, which divided the Brazilian territory among themselves. For example, the Federal University of Pelotas, RS, led a group of federal universities in the states of RS, SC, GO, MG, MA and DF and so it was with the others.

In all cycles, information on prenatal care was obtained through Module III, for women enrolled in the team, aged 18 or over and present at the UBS at the time of the external assessment. Those who answered affirmatively to the following questions: (1) "Have you ever been pregnant?"; (2) If yes: "Do you have children under the age of two?"; (3) If yes: "Did you have prenatal care in the last pregnancy?"; (4) If yes: "Was the prenatal care performed at this health unit?" were included in the analyses.

For the quality assessment, a set of six dichotomous outcomes: 1) six or more prenatal visits; 2) receiving tetanus vaccine (if necessary); 3) prescription for the use of ferrous sulfate; 4) physical examination; 5) educational guidelines and 6) laboratory tests. For physical examination (item 4), we asked about six procedures performed during visits (no, yes): measurement of uterine height and blood pressure, gynecological, breast and oral cavity examinations, and cervical cancer prevention. For educational guidelines (item 5) were evaluated (no, yes): food and weight gain, exclusive breastfeeding up to six months, care for the baby and the importance of the cervical cancer prevention test. For laboratory tests (item 6), the woman was asked

if she had undergone the following tests at least once during pregnancy (no, yes): common urine, blood glucose, anti-HIV, syphilis detection (VDRL) and ultrasound. These indicators used in the analysis were extracted from Low-Risk Prenatal Care guideline, which guided the elaboration of the instruments applied.

To compare samples between cycles, were analyzed the distribution of women by age in years (less than 20, 20 to 24, 25 to 34 and 35 or more), ethnicity according formal classification of the Ministry of Health (white, black, mixed race, indigenous) and whether they were beneficiaries of the "Bolsa Família" Program (yes, no).

The Stata 12.0 statistical package (StataCorp LP, College Station, United States) was used for data analysis. In addition to the descriptive analysis, the prevalence of the six outcomes were calculated for each cycle independently and the results were stratified by region (North, Northeast, Center-West, Southeast and South). Variance-weighted least squares regression was applied to estimate the mean of the absolute annual variation in the prevalence of each indicator, in order to include different time intervals between external assessments and to test the statistical significance of trends observed with a p-value less than or equal to 0.05%. The year in which data were collected was used as an independent variable in time trend analysis.

Studies were approved by the Research Ethics Committee (CEP) of the School of Medicine of the Federal University of Pelotas—38/12 of May 10, 2012 and 2.453.320 of December 27, 2017—for cycles I and III, respectively. Cycle II was approved by CEP of the Federal University of Goiás, under opinion 487.055, of December 2, 2013. All participants signed the Informed Consent Form. The authors declare that there are no conflicts of interest in relation to the topic of study.

## Results

In the first external assessment cycle of PMAQ held in 2012, 8,777 women with children under two years of age were interviewed, 98.7% of whom underwent prenatal care and, of these, 70.7% (n = 6,125) did so at the UBS evaluated. In the second cycle, in 2014, there were 13,043 women with children under two years old, 99.8% of whom underwent prenatal care and, for 76.4% of them, the follow-up was carried out in the evaluated UBS (n = 9,945). In 2018, in the third cycle, there were 21,110 women with children under two years of age, 97.3% of whom underwent prenatal care and, of these, 80.8% did so in the evaluated UBS (n = 16,596).

In cycle 3, compared to cycle 1, after six years, a higher proportion of mothers aged 35 years or more was observed—12% to 15%—and a lower proportion of mothers aged up to 20 years —20% to 16%. There was an increase in the proportion of black or brown mothers—64% to 72%. The receipt of benefits from the "Bolsa Família" Program was lower, from 56% in 2012 to 52% in 2018 (Table 1).

Considering the prevalence of the indicators over the period, a positive evolution was observed for HIV, syphilis, glycemia and ultrasound exams in isolation and for all exams. There was also an increase in the prevalence of receiving guidance on feeding and weight gain of the baby and examination of the oral cavity (Table 2).

On the other hand, we observed reduction in the proportions of having a tetanus vaccine, six or more visits, receiving guidance on exclusive breastfeeding and care for the newborn, and in all procedures that included uterine height measurement, gynecological examination and cervical cancer prevention (Table 2).

Finally, the following components of care for pregnant women have not changed significantly over the period: prescription of ferrous sulfate, common urine test, guidance on the importance of cervical cancer preventive testing, receipt of all guidelines and examination of breasts (Table 2).

**Table 1. Distribution of users interviewed in each cycle according to sociodemographic characteristics.** PMAQ, Brazil, 2012 to 2018.

|  | Cycle 1 | Cycle 2 | Cycle 3 |
|---|---|---|---|
|  | 2012 | 2014 | 2018 |
|  | n = 6,125 | n = 9,945 | n = 16,596 |
| Age (years) |  |  |  |
| Up to 20 | 20.0% | 18.6% | 15.8% |
| 21 to 24 | 24.6% | 23.1% | 22.8% |
| 25 to 34 | 43.6% | 46.1% | 46.9% |
| 35 or older | 11.8% | 12.2% | 14.5% |
| Ethnicity* |  |  |  |
| White | 32.5% | 28.9% | 23.6% |
| Black / Brown / Mixed | 63.6% | 66.6% | 71.8% |
| Indigenous | 3.9% | 4.5% | 4.6% |
| Beneficiary of the "Bolsa Família" Program | 55.9% | 47.2% | 51.6% |

* According the formal classification of the Ministry of Health.

It is necessary to highlight the downward trend in the proportion of women who had the recommended number of, at least, six prenatal visits, from 89% in 2012 to 84% in 2014 and 78% in 2018 (p <0.001) (Table 2).

In spite of the fact that some tests had an almost universal request in 2012—common urine, HIV and ultrasound—and if maintained in subsequent cycles, the growth in the request for

**Table 2. Prevalence of prenatal quality indicators and average annual percentage change.** PMAQ, Brazil, 2012 to 2018.

|  | Cycle 1 2012 | Cycle 2 2014 | Cycle 3 2018 | Average annual change (percent points) (95% CI) | p-value |
|---|---|---|---|---|---|
| Six or more visits | 89.1% | 84.3% | 77.8% | -2.25 (-2.43;-2.08) | <0.001 |
| Tetanus vaccine | 97.0% | 89.2% | 90.4% | -1.22 (-1.34; -1.10) | <0.001 |
| Ferrous sulfate prescription | 96.5% | 96.7% | 95.9% | -0.14 (-0.26;-0.02) | <0.05 |
| **Exams:** |  |  |  |  |  |
| Common urine | 97.7% | 97.6% | 97.6% | -0.02 (-0.09;0.06) | 0.711 |
| HIV | 94.9% | 94.8% | 96.9% | 0.49 (0.38;0.60) | <0.001 |
| Ultrasound | 94.4% | 93.9% | 95.5% | 0.31 (0.19;0.42) | <0.001 |
| Syphilis | 84.3% | 87.5% | 95.3% | 2.34 (2.18–2.50) | <0.001 |
| Glucose | 82.9% | 86.1% | 92.5% | 1.99 (1.82;2.17) | <0.001 |
| **All exams** | **69.2%** | **74.4%** | **86.1%** | **3.54 (3.32;3.76)** | **<0.001** |
| **Guidelines:** |  |  |  |  |  |
| Exclusive breastfeeding | 91.0% | 90.8% | 87.2% | -0.87 (-1.03;-0.71) | <0.001 |
| Food and weight gain | 88.9% | 87.9% | 90.6% | 0.49 (0.33;0.65) | <0.001 |
| Baby care | 85.9% | 85.2% | 83.2% | -0.57 (-0.76;-0.38) | <0.001 |
| Importance of pre-cancer | 65.8% | 65.3% | 65.5% | -0.02 (-0.27;0.22) | 0.845 |
| **All guidelines** | **60.3%** | **59.8%** | **59.3%** | **-0.19 (-0.45;0.07)** | **0.151** |
| **Procedures** |  |  |  |  |  |
| Blood pressure measurement | 98.7% | 98.8% | 99.1% | 0.09 (0.03;0.14) | <0.05 |
| Measurement of uterine height | 97.4% | 96.7% | 94.5% | -0.60 (-0.70;-0.51) | <0.001 |
| Breast Exam | 56.3% | 56.3% | 55.9% | -0.09 (-0.36;0.17) | 0.478 |
| Examination of the oral cavity | 45.0% | 49.2% | 54.7% | 1.91 (1.65;2.17) | <0.001 |
| Gynecological examination | 44.7% | 41.4% | 35.1% | -1.97 (-2.23;-1.71) | <0.001 |
| Pre-cancer screening | 35.8% | 35.3% | 31.3% | -1.02 (-1.27;-0.77) | <0.001 |
| **All procedures** | **17.0%** | **16.7%** | **13.5%** | **-0.80 (-0.99;-0.62)** | **<0.001** |

VDRL and blood glucose in 2014 and 2018 is noteworthy. This improvement had an effect on the synthetic indicator of request for all exams, from 69% in 2012 to 74% in 2014 and 86% in 2018 (p <0.001) (Table 2).

In relation to the guidelines, the slight growth of the guidance on diet and weight gain stands out positively (p <0.001), but the themes of exclusive breastfeeding up to six months and guidance on care for the newborn had a significant decrease (p <0.001). Guiding the pregnant woman about the importance of the cervical cancer prevention tests remained at similar levels (Table 2).

For indicators related to procedures and physical examination, the situation is worrying. Despite the growth in the examination of the oral cavity, the measurement of the uterine height, the gynecological examination and the preventive examination for cervical cancer decreased significantly between 2012 and 2018. Add to that the stability in the proportion of examination of the breasts, the synthetic indicator had a significantly negative evolution (p <0.001) (Table 2).

Regardless of the evolution in the prenatal care indicators, it should be noted that more than 90% pregnant women reported in 2018 the tetanus vaccination (90%), the prescription of ferrous sulfate (96%), having a common urine test (98%), blood glucose (93%), anti-HIV (97%), VDRL (95%) and ultrasound (96%), in addition to having their blood pressure (99%) and uterine height (95%) measured. However, levels below 70% were recorded for guidance on the importance of preventive examination (66%) and for physical examination of the breasts (56%), oral cavity (55%), and gynecological examination (35%) and material collection for pre-cancer (31%) (Table 2).

Stratifying by region, prenatal care indicators showed different scenarios. The proportions of pregnant women who had six or more visits and who received tetanus vaccine fell significantly over the period in all regions. On the other hand, the proportion of pregnant women undergoing all the investigated exams increased, without distinction by region. The prescription of ferrous sulphate increased in the North and Center-West and reduced in the Northeast, Southeast and South regions. The supply of all guidelines showed a tendency for growth in the Northeast and Center-West and a decrease in the Southeast. The performance of all procedures improved in the Northeast, Center-West and South regions, having reduced in the Southeast region. (Table 3).

## Discussion

The study verified the evolution of coverage and quality indicators of prenatal care in primary care in Brazil in the period from 2012 to 2018, in an unprecedented analysis. The first result of this study, related to coverage, was positive with the systematic tendency to increase the proportion of women undergoing prenatal care at UBS throughout the PMAQ cycles, increasing from 71% to 81% in the period. According to the latest edition of the 2019 National Health Survey, 70% Brazilians used SUS when they needed health services [23].

The availability of PHC services offering this programmatic action is wide and this proportion is similar to that which represents the users of the Unified Health System in the population as a whole. The growth of poverty and austerity socioeconomic measures have increased the demand on public services [35, 36]. In contrast, the reformulation of the financing policy for Primary Health Care for municipalities, whose calculation now considers the population registered in each of the health teams regardless of the area of residence, translates into a containment measure, restricting the capacity of the health network and bringing potential harm to universality [37–40].

**Table 3. Prevalence of prenatal quality indicators and average annual change (%) according to the region.** PMAQ, Brazil, 2012 to 2018.

| | Cycle 1 2012 | Cycle 2 2014 | Cycle 3 2018 | Average annual change (%) (percent points) (95% CI) | p-value |
|---|---|---|---|---|---|
| Six or more visits | | | | | |
| North | 81.1% | 79.2% | 76.3% | -0.96 (-1.71;-0.21) | <0.05 |
| Northeast | 87.8% | 86.4% | 77.7% | -2.19 (-2.51;-1.88) | <0.001 |
| Center-West | 88.2% | 72.0% | 76.2% | -2.10 (-2.69;-1.50) | <0.001 |
| Southeast | 90.4% | 85.9% | 77.3% | -2.63 (-2.90;-2.36) | <0.001 |
| South | 93.0% | 86.4% | 82.7% | -2.00 (-2.50;-1.50) | <0.001 |
| Tetanus vaccine | | | | | |
| North | 95.9% | 86.1% | 90.2% | -0.72 (-1.20;-0.24) | <0.05 |
| Northeast | 97.9% | 92.6% | 93.4% | -0.82 (-0.99;-0.65) | <0.001 |
| Center-West | 96.5% | 86.8% | 89.1% | -1.36 (-1.78;-0.94) | <0.001 |
| Southeast | 96.8% | 87.8% | 87.9% | -1.76 (-1.97;-1.55) | <0.001 |
| South | 96.3% | 86.1% | 89.1% | -1.28 (-1.70;-0.85) | <0.001 |
| Ferrous sulfate prescription | | | | | |
| North | 94.9% | 95.6% | 97.7% | 0.63 (0.25;1.00) | <0.05 |
| Northeast | 98.3% | 98.2% | 97.7% | -0.13 (-0.26;-0.10) | <0.05 |
| Center-West | 95.2% | 95.7% | 97.2% | 0.43 (0.10;0.77) | <0.05 |
| Southeast | 96.6% | 96.4% | 94.2% | -0.52 (-0.70;-0.35) | <0.001 |
| South | 92.4% | 93.7% | 91.4% | -0.38 (-0.84;0.09) | <0.113 |
| All exams | | | | | |
| North | 55.5% | 60.8% | 83.1% | 6.33 (5.43;7.22) | <0.001 |
| Northeast | 66.4% | 73.0% | 87.5% | 4.43 (4.06;4.81) | <0.001 |
| Center-West | 75.2% | 75.5% | 85.7% | 2.47 (1.77;3.18) | <0.001 |
| Southeast | 71.2% | 77.3% | 84.8% | 2.66 (2.31;3.02) | <0.001 |
| South | 73.6% | 80.8% | 88.4% | 2.81 (2.13;3.49) | <0.001 |
| All guidelines | | | | | |
| North | 54.9% | 54.4% | 57.4% | 0.71 (-0.25;1.66) | <0.147 |
| Northeast | 61.3% | 61.8% | 65.6% | 0.99 (0.56;1.42) | <0.001 |
| Center-West | 45.1% | 46.3% | 53.6% | 1.90 (1.04;2.76) | <0.001 |
| Southeast | 63.4% | 61.3% | 54.2% | -1.95 (-2.37;-1.53) | <0.001 |
| South | 61.7% | 64.6% | 60.1% | -0.67 (-1.51;0.17) | <0.119 |
| All procedures | | | | | |
| North | 11.3% | 16.7% | 15.0% | 0.12 (-0.45;0.68) | <0.689 |
| Northeast | 16.7% | 23.3% | 21.7% | 0.53 (0.20;0.86) | <0.05 |
| Center-West | 8.7% | 14.6% | 17.6% | 1.60 (1.07;2.13) | <0.001 |
| Southeast | 20.0% | 26.6% | 19.5% | -0.69 (-1.01;-0.38) | <0.001 |
| South | 16.2% | 27.5% | 23.4% | 0.75 (0.08;1.41) | <0.05 |

Guimaraes et al. analyzed data from modules I and II of the second cycle of PMAQ, and reported inadequacy in the infrastructure of the primary care network that performs prenatal care, low adequacy of clinical actions for the quality of care and low management capacity of teams to ensure access and quality of care [41]. Their findings corroborate those presented herein, since data from module III of the three cycles were used, answered by users with children under two years old. According to the model proposed by Donabedian [42] for service evaluation, if the structure and work process of the teams are not adequate, the results, from the point of view of the attention received and perceived by users, will also will fall short of what is desired.

Of the six outcomes investigated, four evolved negatively—six or more visits, tetanus vaccination, ferrous sulphate prescription and physical examination procedures—one did not show any variation—provision of guidelines and one had a positive evolution—all tests were performed.

Prenatal quality indicators showed a significant reduction over the cycles evaluated by PMAQ, with a decrease in the number of visits, the tetanus vaccine and the prescription of ferrous sulfate. When observing the set of laboratory tests, it can be emphasized that there was an increase in the prevalence of their performance.

Despite the increase in prenatal coverage in primary network, it is worrying that it has not improved. The recommended minimum number of six visits declined by more than two percentage points per year, which does not seem to have a reasonable explanation, given that in the period there was no record of changes in the protocol, nor changes in the attributes and functions of the teams regarding it refers to the early captation and access and flows of these pregnant women in the health system. Considering the regions, the same behavior was observed, with the Southeast, Northeast and Center-West regions having the worst performances. In Brazil, regional differences may be due to historical socioeconomic inequalities that affect the health system as a whole [43].

Possibly, the increase in the samples of the teams over the years and their representativeness at the national level made them more similar to the totality of care services. In 2012, in Cycle I, the rules at the time promoted the adhesion of the best teams in the municipalities, resulting in a greater proportion of teams classified as good and excellent [31, 32].

The coverage of tetanus vaccination had an annual reduction in the order of 1.2 percentage points between cycles. In Brazil, the National Immunization Program (PNI), instituted in 1973, regulates the use of tetanus vaccine and other vaccines throughout the national territory and sets as a priority the application of tetanus toxoid (TT) or double adult vaccine (dT) in pregnant women [44].

Brazil eliminated maternal and neonatal tetanus as a public health problem in 2003. One of the main factors contributing to the reduction of cases in the country was the adoption of simple preventive measures, such as the application of vaccines.

Since 2016, vaccine coverage in general has declined by about 10 to 20 percentage points in Brazil [45–47]. For example, the resurgence of measles cases in 2018 and 2019 may be due to the decrease in vaccine coverage [47]. In the case of tetanus vaccination in pregnant women, one can also think of the effect of anti-vaccine movements, present in many countries, leading to a phenomenon called vaccination hesitation, characterized by the delay in accepting or refusing the vaccine, regardless of its availability and the access to health services [45–48].

In addition, the increasing underfunding of SUS may have influenced this result, through the shortage of necessary inputs and the spread of false news on different social media [49].

The prescription of ferrous sulfate was the action that maintained the highest rates in the period, above 95%, despite a slight drop. The North and Center-West regions had a positive variation and the Northeast and Southeast regions, a negative variation. According to the WHO (2013), the importance of reinforcing the maternal nutritional supply is great, with the supplementation of vitamins and minerals necessary to supply this increased demand, one of the most relevant is iron supplementation. Anemia affects about 50% pregnant women in the world and is the most frequent hematological disease in this period, with iron deficiency as the main cause [50, 51].

The increase in the prevalence of syphilis tests during pregnancy in the period can be explained by the greater availability of rapid tests at the health units. In the first cycle of the PMAQ, only 3% health units reported having the test always available and this proportion was 27% and 75% in cycles II and III, respectively. It is also likely that this result may be related to

the improvement in the structure of human resources, with more permanent education initiatives on prenatal care, health education for pregnant women and training for professionals in carrying out the test [52].

On the other hand, the increase in the performance of fasting blood glucose tests seems to indicate a greater concern of professionals with gestational diabetes due to greater adherence to the protocols, in addition to greater access for pregnant women to laboratory diagnostic support services in the municipal health systems.

Contrary to what occurred with the indication of prenatal examinations, the provision of guidance on breastfeeding and care for the baby had a significant drop over the cycles. This result is worrying because the action of providing information does not depend on the availability of inputs or equipment (but it can depend on time, training, knowledge/strategies on how to do it, so that the guidance given remains in people's memory and is remembered when asked). It is believed that one of the factors that can cause professionals to reduce the supply of information during prenatal care is related to the work process of PHC teams. Excessive demand in the service routine, with a consequent reduction in the time of each service, can cause professionals to dismiss the guidelines to a second plan. The possibility of recall bias was also examined, as the pregnant woman might not remember all the orientations received, mainly due to the age of their children. In all cycles, there were no differences in the report of orientations received according to the age of the children, even mothers of younger children reported less guidance in cycles II and III than mothers of children older than six months.

It should be noted the increase in the proportion of examination of the oral cavity throughout the PMAQ cycles. Although this exam can be performed by physicians and nurses to detect needs for follow-up by the dentist, this change may be partly a reflection of the increased coverage of oral health teams in PHC in Brazil. Data from the e-manager system, available at, report that in 2012 the coverage of oral health teams was 38% and in 2018 was 42%. Another possible explanation would be an increase in adherence to the, with a change in the role of the dentist in the FHS teams, which started to be inserted in prevention and promotion activities for all age groups [20].

There was a sharp drop in the proportion of gynecological exams during prenatal care, one of the most worrying findings in the study. This indicator was already low, 45% in 2012 and dropped to 35% in 2018. During the period, there was no shortage of supplies and materials necessary for the procedure; on the contrary, the availability of a gynecological table, gloves and vaginal speculums increased throughout Brazil, reaching almost 100% PHC health units in the third cycle. Thus, we have no other plausible explanation for this reduction in the indicator other than that related to the work process, already raised in the comments on the guidelines. Another possible explanation would be the profile of physicians in PHC, which, due to provision programs such as PROVAB and MAIS DOCTORS, could be younger, with little experience and security to perform the exam, although in the FHS, nurses do prenatal and gynecological exams [53, 54].

Among the limitations of the study, the fact that the samples of women interviewed were located in health units that adhered to the external assessment process of the PMAQ stands out. This strategy may have constituted a selection bias, capable of inflating the positivity of the findings, by disregarding those without regular access to the service [55]. Nevertheless, cycles II and III had the adherence of more than 74% and 93% of the teams, respectively, and the differences in the demographic characteristics observed support the idea that, with the passing of the cycles, the samples tend to resemble more the general population, such as skin color [56]. Another limitation can be attributed to the instruments and their limited number of variables to assess the quality of prenatal care, despite the fact that the same questions were made in the three cycles. Still, the answers could vary according to the recall period, as women

referred to the period of pregnancy and their children could be up to two years old. For this, the data were stratified according to the child's ages, without significant variations.

Among the most positive aspects, the magnitude of the samples and the national scope of their distribution stands out, something unprecedented in Brazil, in addition to the opportunity to compare indicators of care over six years. The centrality of users in external evaluation is another highlight of PMAQ, as it values the perception of the citizen in identifying problems and improving daily practices [16]. The analytical strategy was also suitable for cross-sectional time series designs.

The PMAQ was completed in the federal administration initiated in 2019, which replaced it by the PREVINE Brazil Program [34]. The new program produced a setback in the AB financing paradigm, by focusing the incentive on individual registration and on achieving goals for a small number of selected indicators, despite three of them being related to prenatal care, without definition of territory and population reference, among other issues that structurally affect the ESF model [57]. In the specific case of prenatal care, it is worth remembering that the biggest challenge is not to guarantee six appointments or more, which is already a reality for about 80% of pregnant women, but to perform the gynecological examination during the course of pregnancy, which reached only a third of women interviewed in Cycle III.

## Conclusions

The extensive offer of prenatal care in primary network is not enough to ensure a successful pregnancy. Qualified care could avoid adverse outcomes during pregnancy, childbirth and the puerperium, but it is necessary that managers and health professionals in the network strive to improve captation for early prenatal care, in order to impact the proportion of pregnant women who perform six or more visits and receive tetanus vaccine, two indicators that showed significantly worse performance over the analyzed period. The positive evolution of the request indicator for all exams should be highlighted positively, with such growth being mainly attributable to the increased demand for VDRL and blood glucose tests. It is also necessary to reinforce the guidance on exclusive breastfeeding and regarding newborn care, as both indicators also showed negative performance over the period. Regarding the procedures, those who had the greatest reductions were the gynecological exam, the performance of a preventive exam for cervical cancer and the measurement of uterine height. All of these actions are important to ensure quality prenatal care and are based on consolidated guidelines and protocols. Even so, the application of these actions depends exclusively on the professional practice, without requiring costs for the health system.

## Acknowledgments

Our thanks to the Family Health teams and the users who were interviewed and contributed to this assessment.

## Author Contributions

**Conceptualization:** Elaine Tomasi, Thales Moura de Assis, Paulo Guilherme Muller, Luiz Augusto Facchini.

**Formal analysis:** Elaine Tomasi, Thales Moura de Assis, Paulo Guilherme Muller, Denise Silva da Silveira, Rosália Garcia Neves, Everton Fantinel, Elaine Thumé, Luiz Augusto Facchini.

**Methodology:** Elaine Tomasi, Thales Moura de Assis, Paulo Guilherme Muller, Denise Silva da Silveira, Rosália Garcia Neves, Everton Fantinel, Elaine Thumé, Luiz Augusto Facchini.

**Project administration:** Elaine Tomasi, Luiz Augusto Facchini.

**Writing – original draft:** Elaine Tomasi, Thales Moura de Assis, Paulo Guilherme Muller, Denise Silva da Silveira, Rosália Garcia Neves, Everton Fantinel, Elaine Thumé, Luiz Augusto Facchini.

**Writing – review & editing:** Elaine Tomasi, Thales Moura de Assis, Paulo Guilherme Muller, Denise Silva da Silveira, Rosália Garcia Neves, Everton Fantinel, Elaine Thumé, Luiz Augusto Facchini.

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
