## [Decision Letter · Decision Letter 0]

1 Sep 2021

PONE-D-21-11493

Evolution of the quality of prenatal care in the primary network of Brazil from 2012 to 2018: what can (and should) improve?

PLOS ONE

Dear Dr. TOMASI,

Thank you for submitting your manuscript to PLOS ONE. After careful consideration, we feel that it has merit but does not fully meet PLOS ONE’s publication criteria as it currently stands. Therefore, we invite you to submit a revised version of the manuscript that addresses the points raised during the review process.

The authors present the findings of the Brazilian National Program for the Improvement of Access and Quality in Primary Care (PMAQ) reviews of primary care antenatal services at 3 time points between 2012 and 2018. On-going evaluation of public services to which this analysis contributes is important for improvements in quality of care.

However, this manuscript is conceptually and methodologically unclear and not suitable for publication is its present form.

Please respond to the comments of Reviewers 2 and 3 and to my own suggestions in the attached document which detail some of these concerns.

We look forward to receiving your revised manuscript.

Kind regards,

Emma K. Kalk

Academic Editor

PLOS ONE

Journal Requirements:

Additional Editor Comments (if provided):

See attached document

Reviewers' comments:

Reviewer's Responses to Questions

**Comments to the Author**

1. Is the manuscript technically sound, and do the data support the conclusions?

Reviewer #1: Yes

Reviewer #2: Partly

Reviewer #3: No

2. Has the statistical analysis been performed appropriately and rigorously? 

Reviewer #1: Yes

Reviewer #2: Yes

Reviewer #3: I Don't Know

3. Have the authors made all data underlying the findings in their manuscript fully available?

Reviewer #1: Yes

Reviewer #2: No

Reviewer #3: No

4. Is the manuscript presented in an intelligible fashion and written in standard English?

Reviewer #1: Yes

Reviewer #2: Yes

Reviewer #3: No

5. Review Comments to the Author

Reviewer #1: Manuscript prepared by researchers with experience in the evaluation of services and actions in primary health care, with a significant number of publications with a similar theme. They present a consistent overview of the development of prenatal care over three valid evaluation cycles covering Brazil as a whole.

Justification adequate, consistent and supported by updated literature in this field of knowledge. Methodology adequately described and able to support the presentation of results. The data presented support central statements such as the coexistence between good coverage of prenatal care in Brazil and a worrying reduction in the procedures necessary for the prevention and cure of diseases of high incidence during pregnancy.

The conclusions are supported by the data presented and the limitations of the study are adequately presented.

Reviewer #2: Prezados autores,

Inicialmente, gostaria de parabenizá-los pelo excelente estudo. Venho sugerir alguns pequenos ajustes para melhoria da produção:

Introdução: Incluir as recomendações do PHPN e Rede Cegonha para qualificação do Pré-natal no Brasil. Tornar mais clara a situação do PMAQ (não mais vigente), já que desde dezembro de 2019 foi instituído um novo modelo de financiamento da APS brasileira, o Programa Previne Brasil pela Portaria nº 2.979.

Materiais e Métodos: Quais os critérios de inclusão e exclusão adotados? Também senti falta de mais detalhes sobre a amostragem e recrutamento das mulheres: Qual o local das entrevistas e de que forma foram abordadas? Houve estudo piloto do instrumento de coleta? Como as 41 instituições de ensino superior se organizaram para a coleta de dados? Foi realizado cálculo amostral? Em relação aos aspectos éticos, houve utilização do TALE às menores de 18 anos, além do TCLE para seu responsável legal?

Discussão: Senti falta de discutir alguns dos resultados do estudo ... e de trazer mais comparações dos achados com outros estudos nacionais e internacionais ... Ademais, como os autores entendem as diferenças das Regiões brasileiras nos resultados do pré-natal?

*Em um dos parágrafos da discussão, faz-se necessário melhorar a descrição do novo modelo de financiamento da APS no Brasil. O Programa Previne Brasil realizará repasses financeiros com base em três critérios: a) captação ponderada (cadastros das pessoas), b) pagamento por desempenho (sete indicadores, que são distribuídos em pré-natal, saúde da mulher, saúde da criança e doenças crônicas e avaliados quadrimestralmente) e c) incentivo para ações estratégicas. Em relação ao pagamento por desempenho, o repasse dependerá dos resultados alcançados por cada equipe em um conjunto inicial de sete indicadores, dentre os quais os 3 primeiros são referentes ao pré-natal. Foi mantido o início precoce (até a 20ª semana), realização de 6 ou mais consultas, dos exames para sífilis e HIV e o atendimento odontológico na gestação.

Conclusões: Entendo ser importante acrescentar sugestões de possíveis estratégias que a APS poderia adotar para melhoria da qualidade do pré-natal no Brasil. O que os autores visualizam que ainda poderia ser feito, em termos de gestão interfederativa e de assistência às gestantes brasileiras, para garantia de um pré-natal qualificado?

*O Previne Brasil reforçou essa responsabilização da gestão e profissionais de saúde pelas pessoas que assistem ... Como os trabalhadores da APS poderiam se organizar para melhorar os resultados/alcançar as metas dos indicadores de pré-natal?

Respeitosamente,

Avaliador

Reviewer #3: I think the performance of health care systems is an important topic and country examples like this one should be published. However, overall I found this article underdeveloped conceptually and very difficult to follow.

The stated purpose of the paper is to “describe the temporal evolution of the coverage and quality of prenatal care performed in primary care in Brazil…” but key concepts like quality are not well articulated or defined. There is no clear discussion in the introduction of what the authors mean by quality (at the moment there is a brief mention of quality on line 61 that is followed directly by a reference to access) and then very little in the methods to show how these concepts were reflected in study design, data collection and analysis. A clearer explanation of the PMAQ and the indicators developed is needed to contextualise the study and provide information for the reader.

Regarding the methods, what is the sampling approach and how were participants selected? What proportion of the eligible population does this sample represent in each of the districts and therefore how can we interpret the meaning of these findings?

I am confused by statements like: Line 107 “A total of six dichotomous outcomes (no; yes) were selected” because I am unclear how this statement relates to the previous paragraph which refers to interviews. And again, Line 113 "Four educational guidelines were 113 evaluated (no, yes)"

Line 118 "The indicators used in the analysis were validated by the technical areas of the Ministry of Health. – It would be useful to have more details about this

Sentence beginning Line 119 does not seem to be finished. Needs restating for clarity.

124 - ‘prevalence of outcomes’ was calculated. What does this mean?

157 – “Considering the prevalence of the indicators over the period, a positive evolution was observed for…”

Until the conceptual and methodological issues are clarified, I do not think I can provide useful comments on the discussion. At present, parts of the discussion that refer to explanations for change in indicators over time such as the paragraphs beginning on LIne 286 and Line 319 require references.

6. PLOS authors have the option to publish the peer review history of their article (what does this mean?). If published, this will include your full peer review and any attached files.

Reviewer #1: No

Reviewer #2: **Yes: **José Marcos de Jesus Santos

Reviewer #3: No

---

## [Author Response · Author response to Decision Letter 0]

20 Oct 2021

Response to the academic editor

COMMENT: The overall style is not clear and there are some grammatical errors. This is possibly due to translation into English.

ANSWER: We would like these few problems with the language to be pointed out so that we can fix it.

COMMENT 

Introduction: 

I suggest contextualizing the study in Brazil’s antenatal services and health care structures at the time of the reviews. You have not made clear that the PMAQ is a pay for performance scheme and how this works (also noted by reviewer 2); you should distinguish between the different service packages. I would not previous review findings and existing regional differences.

ANSWER: We added content about PMAQ in the Introduction and also in Methods.

COMMENT 

Methods: 

As noted by reviewers 2 and 3, many methodological details are absent. The methodology is partially described. It would be useful to have a summary or table of the relevant antenatal care guidelines.

While each review may be cross-sectional, the analysis is more longitudinal at the 3 time points.

ANSWER: We added content in Methods.

COMMENT 

Please motivate for the selection of the 6 outcomes and the exclusion of others. At which antenatal visits were the elements of “physical examination”, “education” and “lab tests” required?

ANSWER: We rewritten the methods; there was no mention in the instruments to which prenatal consultations the elements were questioned; the question was, for example: "In the prenatal consultation(s), did the health professionals examine your mouth?"

COMMENT 

117: what does this sentence mean? Is there a reference? “The indicators used in the analysis were validated by the technical areas of the Ministry of Health.”

ANSWER: The text has been modified to make it clearer.

COMMENT 

120: please use an alternative term to ‘skin colour’. Maybe ethnicity? And I suggest dropping the word, ‘yellow’. If these terms are necessary, are they part of the official language of the Dept of Health? If so, I suggest stating that ethnicity was characterised using the formal classification of the Department.

ANSWER: Changes made to the text and table 1

COMMENT 

Results 

Need to include uncertainty ranges for all measures.

Is the Annual change in Table 2 a risk difference? Between Cycles I and III? Is it the β-value of the regression?

ANSWER: Confidence intervals are already presented for the main measure (annual change); the measure is not a risk difference, it is an average rate of change between cycles, the β-value of the regression, and represents the increase or decrease in percentage points; for example, for the first indicator, of six or more appointments, the average rate was negative, there was an average decrease of 2.25 percentage points between 2012 and 2018 in the proportion of women who had six or more medical appointments during pregnancy.

COMMENT 

There a very small changes in proportion, perhaps the p-value is due to the large numbers? Do the you think the changes are clinically or programmatically significant?

ANSWER: The changes reflect the actions taken by the health teams, within the scope of their possibilities and incentives received as a result of the Program. It is believed that the PMAQ was able to induce important improvements in almost all investigated indicators, especially in the teams' work process. This can have both clinical and programmatic impact. Small changes in proportions that were already high were expected

COMMENT 

What has the addition of Cycle III data added to that already published? The changes between cycles II and III are small and maybe the less interesting observations in the study.

ANSWER: The inclusion of data from cycle III marks the end of the program and points to the continued growth that would be expected with the continuity of the PMAQ.

COMMENT 

You have not made clear that the PMAQ is a pay for performance scheme. This certainly limits the generalization of the findings to services outside the PMAQ scheme.

ANSWER: The PMAQ was not a typical pay-for-performance program as the financial resources were not transferred directly to professionals, as in other countries, but were transferred to municipal managers.

Response to the reviewer 2.

COMMENT 

Introdução: Incluir as recomendações do PHPN e Rede Cegonha para qualificação do Pré-natal no Brasil. Tornar mais clara a situação do PMAQ (não mais vigente), já que desde dezembro de 2019 foi instituído um novo modelo de financiamento da APS brasileira, o Programa Previne Brasil pela Portaria nº 2.979.

ANSWER

As recomendações do PNPH e da Rede Cegonha foram incluídas; também foram esclarecidos aspectos sobre o fim do PMAQ e sua substituição pelo Programa Previne Brasil.

COMMENT 

Materiais e Métodos: Quais os critérios de inclusão e exclusão adotados? Também senti falta de mais detalhes sobre a amostragem e recrutamento das mulheres: Qual o local das entrevistas e de que forma foram abordadas? Houve estudo piloto do instrumento de coleta? Como as 41 instituições de ensino superior se organizaram para a coleta de dados? Foi realizado cálculo amostral? Em relação aos aspectos éticos, houve utilização do TALE às menores de 18 anos, além do TCLE para seu responsável legal?

ANSWER: Todas as respostas a estas perguntas foram acrescidas na seção de Métodos.

COMMENT 

Discussão: Senti falta de discutir alguns dos resultados do estudo ... e de trazer mais comparações dos achados com outros estudos nacionais e internacionais ... Ademais, como os autores entendem as diferenças das Regiões brasileiras nos resultados do pré-natal?

ANSWER: Não foram localizados resultados comparáveis em outros estudos nacionais e internacionais. As diferenças regionais são decorrentes de históricas desigualdades socioeconômicas que afetam o sistema de saúde como um todo. Foram acrescentadas referências para estes resultados.

COMMENT

*Em um dos parágrafos da discussão, faz-se necessário melhorar a descrição do novo modelo de financiamento da APS no Brasil. O Programa Previne Brasil realizará repasses financeiros com base em três critérios: a) captação ponderada (cadastros das pessoas), b) pagamento por desempenho (sete indicadores, que são distribuídos em pré-natal, saúde da mulher, saúde da criança e doenças crônicas e avaliados quadrimestralmente) e c) incentivo para ações estratégicas. Em relação ao pagamento por desempenho, o repasse dependerá dos resultados alcançados por cada equipe em um conjunto inicial de sete indicadores, dentre os quais os 3 primeiros são referentes ao pré-natal. Foi mantido o início precoce (até a 20ª semana), realização de 6 ou mais consultas, dos exames para sífilis e HIV e o atendimento odontológico na gestação.

ANSWER: Foi melhorada a descrição do Programa Previne.

COMMENT

Conclusões: Entendo ser importante acrescentar sugestões de possíveis estratégias que a APS poderia adotar para melhoria da qualidade do pré-natal no Brasil. O que os autores visualizam que ainda poderia ser feito, em termos de gestão interfederativa e de assistência às gestantes brasileiras, para garantia de um pré-natal qualificado?

ANSWER: Nossas conclusões já apontam estas sugestões.

Response to the reviewer 3

COMMENT 

The stated purpose of the paper is to “describe the temporal evolution of the coverage and quality of prenatal care performed in primary care in Brazil…” but key concepts like quality are not well articulated or defined. There is no clear discussion in the introduction of what the authors mean by quality (at the moment there is a brief mention of quality on line 61 that is followed directly by a reference to access) and then very little in the methods to show how these concepts were reflected in study design, data collection and analysis. A clearer explanation of the PMAQ and the indicators developed is needed to contextualise the study and provide information for the reader.

ANSWER: We've added arguments in the introduction and methods to fulfill this request. References for this conceptual contribution were also added.

COMMENT 

Regarding the methods, what is the sampling approach and how were participants selected? What proportion of the eligible population does this sample represent in each of the districts and therefore how can we interpret the meaning of these findings?

ANSWER: We've added answers to these questions in the Methods section.

COMMENT 

I am confused by statements like: Line 107 “A total of six dichotomous outcomes (no; yes) were selected” because I am unclear how this statement relates to the previous paragraph which refers to interviews. And again, Line 113 "Four educational guidelines were 113 evaluated (no, yes)".

ANSWER: A new wording was made to clarify these doubts.

COMMENT 

Line 118 "The indicators used in the analysis were validated by the technical areas of the Ministry of Health. – It would be useful to have more details about this.

ANSWER: The sentence that generated the doubt was deleted and other explanations were added.

COMMENT 

Sentence beginning Line 119 does not seem to be finished. Needs restating for clarity.

ANSWER: Corrected

COMMENT 

124 - ‘prevalence of outcomes’ was calculated. What does this mean?

ANSWER: A new wording was made to clarify these doubts.

COMMENT 

Until the conceptual and methodological issues are clarified, I do not think I can provide useful comments on the discussion. At present, parts of the discussion that refer to explanations for change in indicators over time such as the paragraphs beginning on LIne 286 and Line 319 require references.

ANSWER: References have been added in both paragraphs.

---

## [Decision Letter · Decision Letter 1]

15 Nov 2021

PONE-D-21-11493R1Evolution of the quality of prenatal care in the primary network of Brazil from 2012 to 2018: what can (and should) improve?PLOS ONE

Dear Dr. TOMASI,

Thank you for submitting your manuscript to PLOS ONE. After careful consideration, we feel that it has merit but does not fully meet PLOS ONE’s publication criteria as it currently stands. Therefore, we invite you to submit a revised version of the manuscript that addresses the points raised during the review process. Thank you for your revisions; the manuscript is now much clearer. Minor queries:line 140: what is TALE? The abbreviation has not been spelled out or explained.line 159 and Table 1: please delete the descriptor, 'yellow'. Indigenous is sufficiently clear.Tables: please indicate that 'Annual % change' is *average* annual % change.

We look forward to receiving your revised manuscript.

Kind regards,

Emma K. Kalk

Academic Editor

PLOS ONE

Journal Requirements:

Reviewers' comments:

Reviewer's Responses to Questions

**Comments to the Author**

1. If the authors have adequately addressed your comments raised in a previous round of review and you feel that this manuscript is now acceptable for publication, you may indicate that here to bypass the “Comments to the Author” section, enter your conflict of interest statement in the “Confidential to Editor” section, and submit your "Accept" recommendation.

Reviewer #2: All comments have been addressed

2. Is the manuscript technically sound, and do the data support the conclusions?

Reviewer #2: Yes

3. Has the statistical analysis been performed appropriately and rigorously? 

Reviewer #2: Yes

4. Have the authors made all data underlying the findings in their manuscript fully available?

Reviewer #2: No

5. Is the manuscript presented in an intelligible fashion and written in standard English?

Reviewer #2: Yes

6. Review Comments to the Author

Reviewer #2: Prezados autores,

As considerações da minha revisão foram atendidas, satisfatoriamente. As recomendações do PHPN e da Rede Cegonha foram incluídas na Introdução, além de esclarecidos aspectos sobre o fim do PMAQ e sua substituição pelo atual Programa Previne Brasil. A seção Métodos trouxe maior detalhamento quanto aos procedimentos realizados, e a Discussão teve o acréscimo de outros estudos/achados sobre a temática. Frente ao exposto, sou favorável à aprovação/aceite do manuscrito na versão atual.

Parabéns pelo estudo!

7. PLOS authors have the option to publish the peer review history of their article (what does this mean?). If published, this will include your full peer review and any attached files.

Reviewer #2: **Yes: **José Marcos de Jesus Santos

---

## [Author Response · Author response to Decision Letter 1]

19 Dec 2021

Dear Dr.Emma Kalk

Acadêmica editor / PLOS ONE 

In relation to the manuscript PONE-D-21-11493, on behalf of the authors, I send the answers to each questioned point in the Minor Queries, in order to qualify our manuscript.

COMMENT: line 140: what is TALE? The abbreviation has not been spelled out or explained.

ANSWER: TALE means free and informed consent term; is the term corresponding to the consent form for minors under 18; the sentence was dropped as it was not really necessary. Information about informed consent is already in the last paragraph of the Methods section: “All participants signed the Informed Consent Form”.

COMMENT: line 159 and Table 1: please delete the descriptor, 'yellow'. Indigenous is sufficiently clear.

ANSWER: The descriptor 'yellow' was deleted in the text and Table 1.

COMMENT: Tables: please indicate that 'Annual % change' is average annual % change.

ANSWER: In the tables 2 and 3 the expression was corrected to “average annual % change”

Best regards

Elaine Tomasi

---

## [Editor Report · Decision Letter 2]

20 Dec 2021

Evolution of the quality of prenatal care in the primary network of Brazil from 2012 to 2018: what can (and should) improve?

PONE-D-21-11493R2

Dear Dr. TOMASI,

We’re pleased to inform you that your manuscript has been judged scientifically suitable for publication and will be formally accepted for publication once it meets all outstanding technical requirements.

Kind regards,

Emma K. Kalk

Academic Editor

PLOS ONE
---

## [Editor Report · Acceptance letter]

7 Jan 2022

PONE-D-21-11493R2 

Evolution of the quality of prenatal care in the primary network of Brazil from 2012 to 2018: what can (and should) improve? 

Dear Dr. Tomasi:

I'm pleased to inform you that your manuscript has been deemed suitable for publication in PLOS ONE. Congratulations! Your manuscript is now with our production department. 

Kind regards, 

on behalf of

Dr. Emma K. Kalk 

Academic Editor

PLOS ONE